# The Role of Echocardiography in the Diagnosis of Cardiac Involvement in a Rare Systemic Condition: The Carcinoid Heart Disease: A Case Report and Review of Literature

**DOI:** 10.3390/diagnostics12122929

**Published:** 2022-11-24

**Authors:** Adela Șerban, Alexandra Dădârlat-Pop, Raluca Tomoaia, Claudia Hagiu, Dan Axente, Valer Donca, Mihai Suceveanu

**Affiliations:** 1Cardiology Department, Heart Institute Niculae Stăncioiu, 19-21 Motilor Street, 400001 Cluj-Napoca, Romania; 25th Department of Internal Medicine, Faculty of Medicine, Iuliu Haţieganu University of Medicine and Pharmacy, 8 Victor Babes Street, 400012 Cluj-Napoca, Romania; 3Clinical Rehabilitation Hospital, 46-50 Viilor Street, 400347 Cluj-Napoca, Romania; 43rd Medical Department, “Iuliu Hatieganu” University of Medicine and Pharmacy, 400347 Cluj-Napoca, Romania; 5“Prof. Dr. Octavian Fodor” Regional Gastroenterology-Hepatology Institute, 400394 Cluj-Napoca, Romania; 6Cluj-Napoca Municipal Clinical Hospital, 400139 Cluj-Napoca, Romania

**Keywords:** carcinoid heart disease, valvular heart disease, carcinoid syndrome, echocardiographic diagnosis

## Abstract

Carcinoid heart disease is a rare presentation of the carcinoid syndrome, which is caused by excessive tumoral hormone production and the abundant release of vasoactive substances with systemic expressions. A 62-year-old woman presented with flushing, diarrhea, weight loss, and right-sided heart failure symptoms. Specific carcinoid heart disease features were identified using transthoracic and transesophageal echocardiography at the tricuspid and pulmonic valves. Biomarkers, 99mTc-Tektrotyd scintigraphy, SPECT-CT, and a biopsy later confirmed the diagnosis, and the patient began treatment for the underlying condition.

## 1. Introduction

Carcinoid tumors are rare, with an incidence of 1.2–2.1/100,000 people/year [1]. Most of this type of tumor arises from the gastrointestinal tract (midgut carcinoids) and bronchus (foregut carcinoids). In about 20–30% of the cases, the initial presentation of a carcinoid tumor is the carcinoid syndrome, a result of excessive hormone production. The most frequent manifestations of the carcinoid syndrome are vasomotor changes (flushing), gastrointestinal hypermotility (secretory diarrhea), bronchospasm, and hypotension. These symptoms are caused by the release of vasoactive substances such as serotonin (5-hydroxytryptamine), 5-hydroxytryptophan, histamine, bradykinin, tachykinins, and prostaglandins [2]. 

Carcinoid heart disease (CaHD) can be diagnosed in up to 50% of patients with carcinoid syndrome and may be the initial presentation of the disease in as many as 20% of the patients. CaHD is characterized by pathognomonic plaque-like deposits of fibrous tissue. These deposits appear most frequently on the endocardium of valvular cusps and leaflets, papillary muscles and cords, cardiac chambers, and occasionally on the intima of the pulmonary arteries or aorta. The affected cardiac valves in carcinoid heart disease have thickened leaflets and subvalvular apparatus with fused and shortened chordae and thickened papillary muscles. Most commonly, the CaHD affects the valves and endocardium of the right side of the heart. This is explained by the fact that the left-sided valves are spared due to the inactivation of humoral substances by the lung [2,3].

Carcinoid plaques are present on both of the endocardial surfaces of the tricuspid leaflets, with changes ranging from mild disease (stiff, thickened leaflets with trivial or mild tricuspid regurgitation) to severe disease (fixed, retracted leaflets with severe tricuspid regurgitation with or without accompanying mild or moderate tricuspid stenosis). Carcinoid plaques also frequently affect the pulmonic valve, causing regurgitation, stenosis, or both [4,5].

The mean survival without treatment in patients with malignant carcinoid syndrome ranges from 12 to 38 months from the onset of systemic symptoms [2,6]. CaHD with advanced symptoms (New York Heart Association classes III or IV) has a particularly poor prognosis, and the mean survival is only 11 months, with most patients dying in under 1 year because of progressive heart failure [7]. These aspects make the timing of the diagnosis and treatment initiation especially important in improving the patients’ survival. By presenting this case report, we aim at raising awareness regarding the early and evolving echocardiographic abnormalities CaHD that would make more patients benefit from optimal diagnostic and therapeutic timing and eventually improve their survival.

## 2. Case Presentation

A 62-year-old woman with recently diagnosed severe tricuspid regurgitation, a small ostium secundum atrial septal defect, an atrial septal aneurysm, a small subaortic ventricular septal defect, left bundle-branch block, and right-sided heart failure presented with the presumed diagnosis of an Ebstein anomaly. 

The patient had experienced a progressive increase in bowel movement in the past 6 months, presenting with up to eight stools per day prior to the admission. Over this period, the patient had undergone several gastroenterology consultations, but no organic cause of the accelerated bowel movement had been identified. In addition, during this time, the patient had started to experience shortness of breath and fatigue at progressively lower levels of exercise and had lost around 25 kg. 

The physical exam showed normal blood pressure (130/80 mmHg) and heart rate (80/min), rhythmic heartbeats, with a moderate tricuspid regurgitation murmur, a loud P2, slightly decreased blood oxygen saturation (93% in a.a.), skin flushing around the cheekbones and the upper torso, hepatomegaly (4 cm below the costal margin), and no other pathological findings. 

ECG observed the LBBB with no other remarkable features. Mild hepatocytolysis, slightly raised serum creatinine levels, and elevated NT-pro-BNP were found. The chest X-ray was within limits. The patient was administered diuretic treatment (Furosemide 40 mg/day, Spironolactone 25 mg/day) given the right-sided heart failure.

A transthoracic echocardiography (TTE) was performed with a Phillips EPIQ 7 echo system using a 1.5 MHz probe. It showed a retracted and thickened tricuspid valve with reduced mobility during both opening and closure (Figure 1), severe tricuspid regurgitation (vena contracta—18 mm), moderate tricuspid stenosis (Figure 2a,b), dilated right heart chambers with preserved systolic function, moderate pulmonary regurgitation (Figure 3), and mild pericardial effusion. The insertion of the septal leaflet of the tricuspid valve was identified as being below the insertion of the mitral valve (7.5 mm/m^2^—a borderline value) but not fulfilling the Ebstein anomaly criteria (≥8 mm/m^2^). In order to obtain a more reliable measurement, a transesophageal echocardiography (TOE) was performed with the same ultrasound machine. It confirmed the structural abnormalities described by TTE and the previous measurements, thus excluding the Ebstein anomaly diagnosis. The TOE better showed the infiltrated aspect of the tricuspid and pulmonary valves and their restricted mobility, which, when integrated with the patient history and the physical examination findings, indicated a high probability of a carcinoid tumor with its consecutive carcinoid syndrome. The TOE also observed the previously known atrial septal aneurysm (Figure 4), with a patent foramen ovale and not an ostium secundum atrial septal defect, as well as a restrictive aortic subvalvular ventricular septal defect. A cardiac MRI also reaffirmed the TOE findings (Figure 5).

The diagnosis was confirmed by very elevated levels of 5-hidroxi-indolacetic acid (>60 mg/24 h), chromogranin A (763 μg/L), and serum serotonin (>1000 μg/L). A 99mTc- Tektrotyd scintigraphy and SPECT-CT were performed in order to identify the site of the carcinoid tumor. It showed multiple metastases in the supraclavicular, mediastinal, and retroperitoneal lymph nodes, as well as liver and bone metastases, without distinguishing the primary lesion (Figure 6). 

The patient underwent a surgical biopsy of the supraclavicular lymph nodes. The pathology report described monomorphic cells with hyperchromatic, round-oval nuclei and pale eosinophilic cytoplasm—see Figure 7a. The immunohistochemistry showed intense and diffusely positive chromogranin, CDX2, CK19, mildly positive synaptophysin, and membrane-positive CK56. The above-mentioned findings were suggestive of a G2 moderate-grade neuroendocrine tumor (Figure 7b).

The patient’s clinical condition had severely declined over time. Specific treatment with Lanreotide 120 mg s.c. was initiated, but the patient passed away just 3 months after the echocardiography raised the suspicion of a carcinoid tumor.

## 3. Discussion

Oncologic diseases generally have rapid progression and a poor prognosis, making the importance of timely diagnosis and treatment essential. This is also the case with malignant carcinoid tumors, which can rarely first reveal themselves as carcinoid heart disease, making the diagnosis even more challenging and having a detrimental effect on the patient’s survival. This case highlights the additional difficulties that can appear in the diagnostic process, when a concomitant disease is also suspected. 

The patient was first diagnosed with a possible Ebstein anomaly, which is a relatively rare congenital cardiac malformation characterized by an antero-caudal displacement of the TV annulus and atrialization of the RV, leading to right heart dysfunction and tricuspid regurgitation [8]. The condition is also commonly associated with patent foramen ovale (PFO) and a ventricular septal defect, which were both identified in our patient. Supplementary imaging techniques, TOE and CMR, have definitely excluded the diagnosis. Ebstein anomaly is rarely diagnosed in adulthood, with symptoms appearing at younger ages and the most common adult presentation being that of an unidentified arrhythmia, which was not the case in our patient [9]. To the best of our knowledge, only one case of an associated CaHD and Ebstein anomaly has been reported in the literature [10].

As stated before, echocardiographic features of advanced CaHD include thickening and retraction of immobile tricuspid valve leaflets with consecutive tricuspid regurgitation. Infrequently, tricuspid valve stenosis is observed. A report by Pellikka PA et al. [11] noted the frequency with which these and other findings occurred in 74 patients with CaHD in whom detailed echocardiography was conducted: tricuspid regurgitation (100%, which was moderate to severe in 90% of the cases), pulmonic stenosis (53%), pulmonic regurgitation (81%), left-sided valvular involvement (7%), and small pericardial effusion (14%). As seen in this report, left-sided valve involvement (defined by diffuse thickening of valve leaflets) is less frequent and usually less severe than right-sided valve disease because of vasoactive peptide inactivation within the pulmonary circulation. Left-sided valve disease can be precipitated by right-to-left shunting, or by high levels of circulating vasoactive substances [2,12]. However, this is not consistent, as demonstrated by our patient, who had a patent foramen ovale and a ventricular septal defect but no left-sided valvular heart disease.

In subjects with established CaHD, echocardiography is recommended when the clinical status is significantly changing or every 3–6 months. To assess the severity of CaHD, echocardiographic scoring systems were developed, with the Westberg and Denney scores being the most reliable [13,14].

Therapies that reduce circulating serotonin levels (somatostatin analogs, antitumor therapy) have not been shown to reverse valve disease. Surgical valve replacement is the current standard of care when intervention is indicated, and it represents the only effective therapy for CaHD. It should be considered for symptomatic patients with severe valve impairment and well-controlled metastatic carcinoid disease and symptoms [2,14].

## 4. Conclusions

In conclusion, echocardiography can be an important tool in the timely diagnosis of CaHD and carcinoid syndrome. Its reliability and affordability warrant a reevaluation of its place in the diagnostic protocol for CaHD, to the detriment of more expensive and less widespread investigations.

## Figures and Tables

**Figure 1 diagnostics-12-02929-f001:**
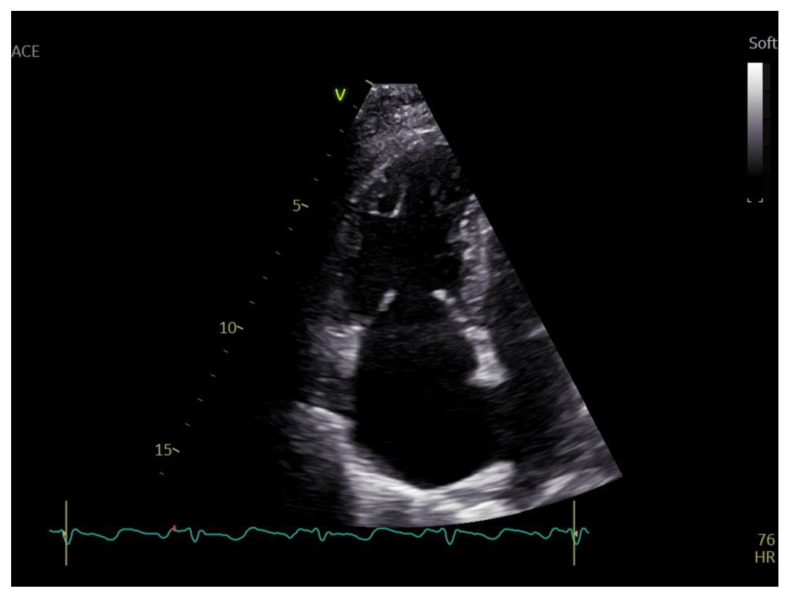
Retracted and thickened tricuspid valve with reduced mobility.

**Figure 2 diagnostics-12-02929-f002:**
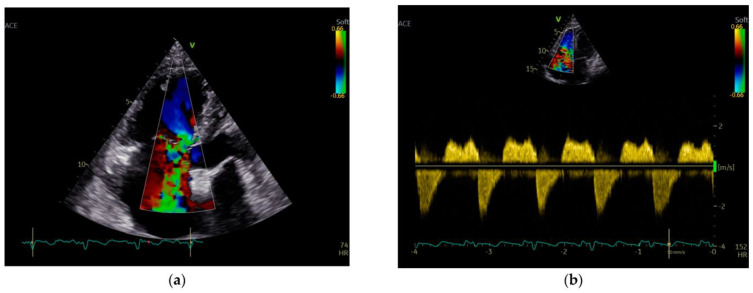
(**a**) Severe tricuspid regurgitation; (**b**) Severe tricuspid regurgitation and moderate tricuspid stenosis.

**Figure 3 diagnostics-12-02929-f003:**
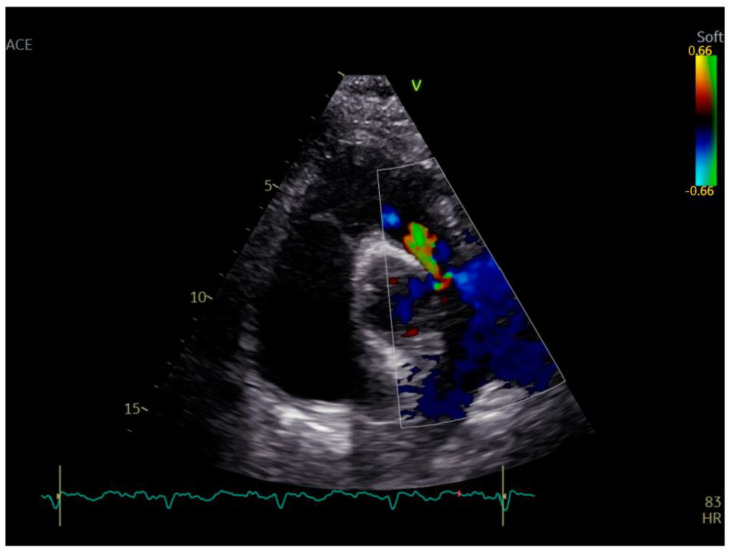
Moderate pulmonary regurgitation.

**Figure 4 diagnostics-12-02929-f004:**
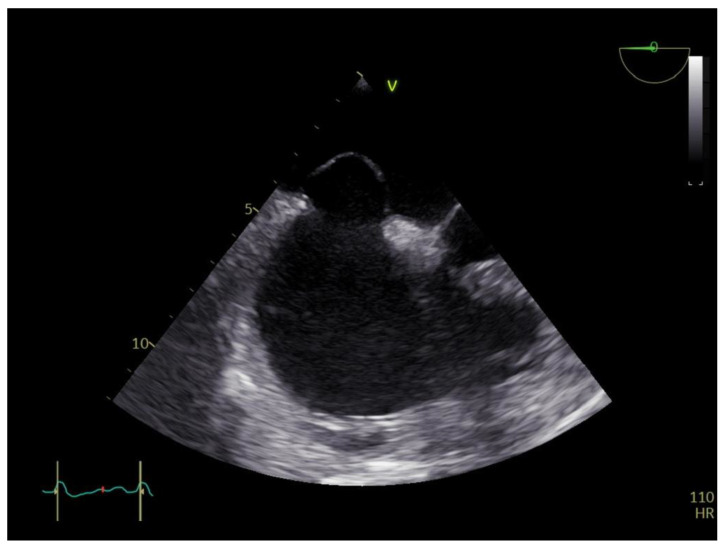
Atrial septal aneurysm.

**Figure 5 diagnostics-12-02929-f005:**
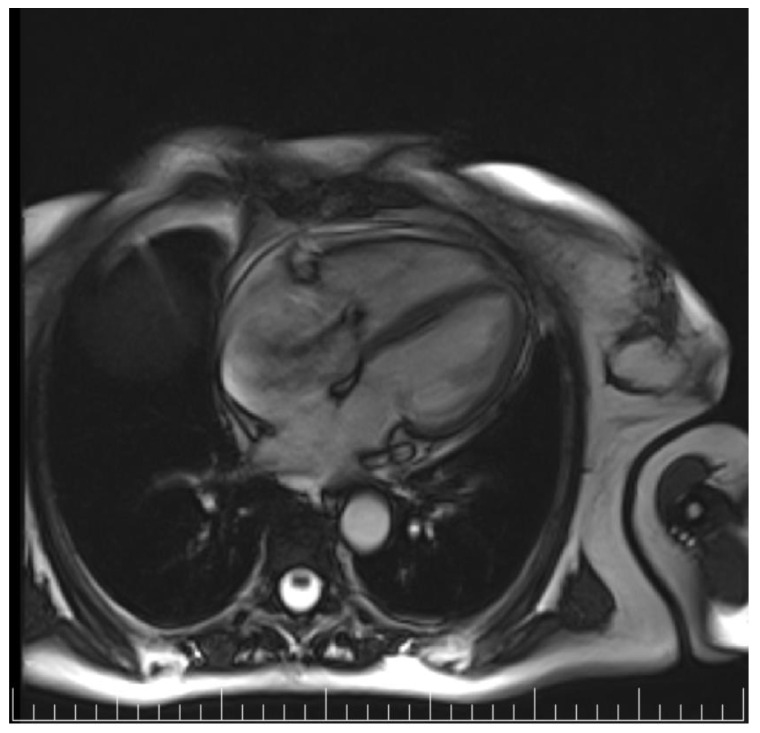
Cardiac MRI—severe tricuspid regurgitation.

**Figure 6 diagnostics-12-02929-f006:**
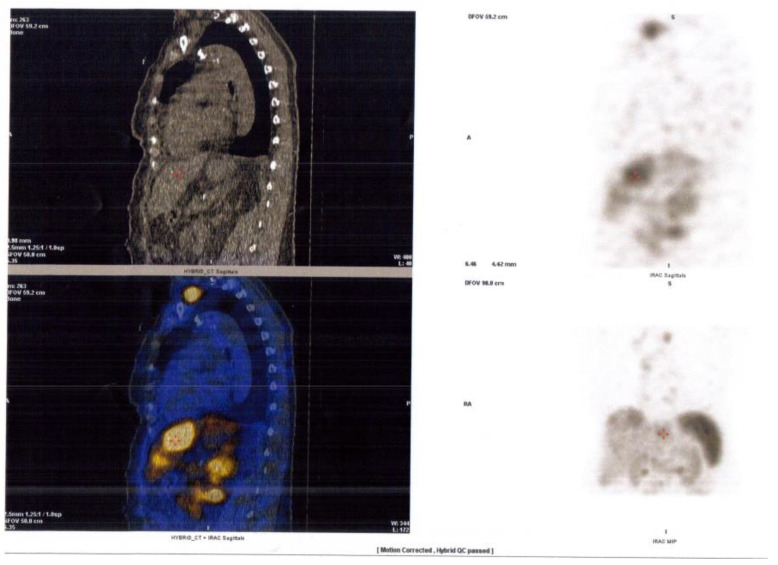
99mTc-Tektrotyd scintigraphy and SPECT-CT showing multiple metastases in the supraclavicular, mediastinal, and retroperitoneal lymph nodes, as well as liver and bone metastases.

**Figure 7 diagnostics-12-02929-f007:**
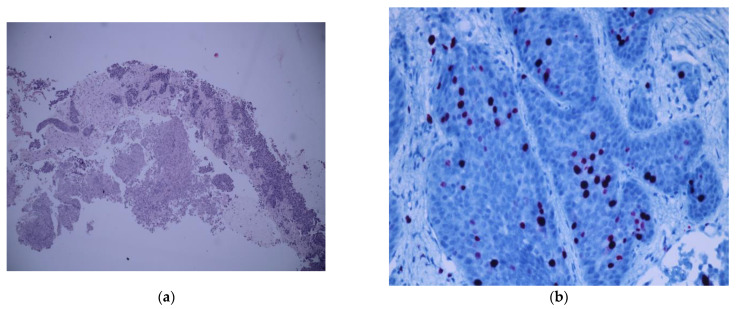
(**a**) Hematoxylin and eosin staining showing numerous small monomorphic cells with salt-and-pepper chromatin; (**b**) immunohistochemical staining for Ki67 labeling index.

## Data Availability

Not applicable.

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
