# Peer review of "The Role of Echocardiography in the Diagnosis of Cardiac Involvement in a Rare Systemic Condition: The Carcinoid Heart Disease: A Case Report and Review of Literature"

_diagnostics, 2022, doi:10.3390/diagnostics12122929_

Round 1

Reviewer 1 Report

In this manuscript, the authors provided a case report of Carcinoid Heart Disease, a rare presentation of the Carcinoid Syndrome caused by excessive tumoral production of hormones and abundant release of vasoactive substances that have systemic expression. The symptoms were identified by Transthoracic and transesophageal echocardiography, and confirmed by  biomarkers, 99m-Tc Tektrotyde scintigraphy, SPECT-CT and biopsy. I have no more comments but one suggestion that the authors should provide the pathological results.

Author Response

We sincerely thank you for the constructive criticism and valuable comments, which were of great help in revising the manuscript. Accordingly, the revised manuscript has been systematically improved with new information and additional interpretations. Also, the article was checked by a professional English translator and there are several changes in terms of spelling and phrasing patterns, as you can observe in the revised paper. We responded to all your comments as follows:

Comment: The authors should provide the pathological result of the biopsy.

Response: Thank you for the suggestion.You can find the new data in yellow in the revised manuscript. The pathology report from the biopsy described monomorphic cells with hyperchromatic round-oval nuclei and pale eosinophilic cytoplasm. The immunohistochemistry showed intense and diffuse positive chromogranin, CDX2, CK19, mild positive synaptophysin and mem-brane positive CK56. The ki-67 proliferation index was below 5%. The above-mentioned findings were suggestive of a G2 moderate grade neuroendocrine tumor.

Reviewer 2 Report

Title: The role of echocardiography in the diagnosis of cardiac 2 involvements in a rare systemic condition: The Carcinoid Heart 3 Disease: A Case Report and Review of Literature

In this manuscript, described carcinoid heart disease (HD), which is occurs when large amounts of vasoactive substances such as serotonin, tachykinins, and prostaglandins reach the right side of the heart, consequent to reduced hepatic metabolism from extensive metastatic liver involvement of the carcinoid tumor.

Overall, the manuscript addresses an un-novelty topic (that is, the pathophysiological drivers of prostate cancer) and stresses the issue of limited therapeutic options.

I have several comments. The manuscript, even though including recent and relevant literature, it should be structured more clearly and follow a line of thought.

1.      Clinical examination of the patient should be described in the case presentation section, like (blood pressure of, atrial fibrillation (AF), jugular venous pressure, cardiomegaly, loud P2, hepatomegaly and clear lung fields).

2.      The presentation of cardiac disease is usually insidious and subtle. Most patients are initially asymptomatic, and clinical features appear along with right-sided heart involvement. Describe and write clearly about it.

3.      In some study mentioned that echocardiogram fails to detect LVNC morphology/hypertrabeculation in a significant number of a cohort of patients with LVNC on CMRI. How will the authors describe LVNC may be missed if echocardiogram is the only imaging modality performed in a cardiac evaluation?

Author Response

Reviewer 2

Comment 1: Clinical examination of the patient should be described in the case presentation section, like (blood pressure of, atrial fibrillation (AF), jugular venous pressure, cardiomegaly, loud P2, hepatomegaly and clear lung fields).

Response 1: We sincerely thank you for the constructive criticism and valuable comments, which were of great help in revising the manuscript. Accordingly, the revised manuscript has been systematically improved with new information and additional interpretations. The article is a Case Report with a short review of existing data- few cases described in the literature. We have made several modifications, all marked in yellow. Also, we believe that our article follows a logical line of reasoning and presents persuasive interpretations and conclusions.

Comment 2: The presentation of cardiac disease is usually insidious and subtle. Most patients are initially asymptomatic, and clinical features appear along with right-sided heart involvement. Describe and write clearly about it.

Response 2:  Thank you for the suggestion. You can find the new data in yellow in the revised manuscript. The patient had experienced a progressive increase in bowel movement in the past 6 months, presenting up to 8 stools per day prior to the admission. Over this period the patient had performed several gastroenterology consultations but no organic cause of the accelerated bowel movement had been identified. In addition, during this time, the patient had started to experience shortness of breath and fatigue at progressively lower levels of exercise and had lost around 25 kg.

Comment 3:    In some study mentioned that echocardiogram fails to detect LVNC morphology/hypertrabeculation in a significant number of a cohort of patients with LVNC on CMRI. How will the authors describe LVNC may be missed if echocardiogram is the only imaging modality performed in a cardiac evaluation?

Response 3:  The left ventricle has no non-compaction morphology. The right ventricle is dilated due to severe tricuspid regurgitation. But, if there is any echocardiographic suspicion of LVNC, CMRI is highly recommended.

We believe that the added data sets the information out more clearly. The changes in the text appear in yellow in the revised paper. We are open to consider further revisions.

Alexandra Dadarlat-Pop
